EMBO
Molecular Medicine

# mRNA mediates passive vaccination against infectious agents, toxins, and tumors

Moritz Thran[1], Jean Mukherjee[2], Marion Pönisch[1], Katja Fiedler[1], Andreas Thess[1], Barbara L Mui[3], Michael J Hope[3], Ying K Tam[3], Nigel Horscroft[1], Regina Heidenreich[1], Mariola Fotin-Mleczek[1], Charles B Shoemaker[2] & Thomas Schlake[1],*

## Abstract

The delivery of genetic information has emerged as a valid therapeutic approach. Various reports have demonstrated that mRNA, besides its remarkable potential as vaccine, can also promote expression without inducing an adverse immune response against the encoded protein. In the current study, we set out to explore whether our technology based on chemically unmodified mRNA is suitable for passive immunization. To this end, various antibodies using different designs were expressed and characterized *in vitro* and *in vivo* in the fields of viral infections, toxin exposure, and cancer immunotherapies. Single injections of mRNA–lipid nanoparticle (LNP) were sufficient to establish rapid, strong, and long-lasting serum antibody titers *in vivo*, thereby enabling both prophylactic and therapeutic protection against lethal rabies infection or botulinum intoxication. Moreover, therapeutic mRNA-mediated antibody expression allowed mice to survive an otherwise lethal tumor challenge. In conclusion, the present study demonstrates the utility of formulated mRNA as a potent novel technology for passive immunization.

**Keywords** antibody; lipid nanoparticle; messenger RNA; mouse; passive immunization
**Subject Categories** Immunology; Microbiology, Virology & Host Pathogen Interaction

## Introduction

Today, vaccination is among the most effective medical treatments for humans and animals, saving millions of lives by inducing immunity against various pathogens. Protection afforded by currently licensed vaccines is primarily based on induction of a humoral immune response (Amanna & Slifka, 2011). It has been demonstrated that neutralizing antibodies are sufficient to prevent dissemination of many diseases within populations (Plotkin, 2010). Hence, passive immunization, by transferring either serum or purified antibodies, is a potent means to confer immediate protection against various threats. Clinical benefit was demonstrated for invasive bacterial infections (Casadevall & Scharff, 1994), for various viral diseases (Janeway, 1945; Hammon *et al*, 1953), as well as for anti-fungal therapy (Bugli *et al*, 2013), and polyclonal antibody products were licensed for several viruses.

With the advent of antibiotics and vaccines, serum therapy was almost abandoned but retained as a niche treatment for toxins, venoms, and distinct viral infections. For instance, post-exposure treatment of rabies as recommended by the World Health Organization consists of the immediate treatment with anti-rabies immunoglobulins in combination with a rabies vaccine (Steele, 1988). As further example, botulism is treated with immune serum (Pediatrics, 1997). Antibody therapies were revolutionized by the development of the hybridoma technology for the production of monoclonal antibodies (mAbs) (Kohler & Milstein, 1975). Since then, more than thirty mAbs have been licensed, but only three of these mAbs are for infectious disease indications, respiratory syncytial virus, anthrax, and very recently rabies in India (Nagarajan *et al*, 2014). Nevertheless, there is ample evidence that it is possible to generate protective mAbs against many viruses and microorganisms (Teitelbaum *et al*, 1998; Nosanchuk *et al*, 2003; Both *et al*, 2013).

Passive immunization declined as a treatment option for a variety of reasons. The primary reason has been the emergence of antibiotics and vaccines. In addition, both serum- and mAb-based therapeutics are costly to produce, store, and deliver as they require a cold supply chain and are often administered intravenously (Chames *et al*, 2009; Keizer *et al*, 2010). Moreover, passive immunization confers only a relatively short duration of protection compared to vaccines. However, there is a renewed interest in passive immunization. Emergence of microbial resistance to antibiotics has increased the demand for alternative therapies. Further, the discovery of broadly neutralizing antibodies may offer new therapeutic options for influenza and HIV (Burton *et al*, 2012). Finally, in contrast to vaccines, antibodies enable a rapid

1  CureVac AG, Tübingen, Germany
2  Department of Infectious Disease and Global Health, Tufts Cummings School of Veterinary Medicine, North Grafton, MA, USA
3  Acuitas Therapeutics, Vancouver, BC, Canada
   *Corresponding author. Tel: +49 7071 9883 1607; E-mail: thomas.schlake@curevac.com

onset of protective immunity. Particularly in a pandemic, passive immunization can offer a critical advantage for blocking virus spread.

As an alternative to the administration of recombinant antibodies, DNA-based approaches have been extensively investigated. Both plasmids as well as viral vectors (adenovirus, adeno-associated virus) have been used for passive immunization. Using viral vectors, sustained expression of low to mid microgram per milliliter levels was obtained in different models and conferred protection against influenza and RSV challenges (Lewis *et al*, 2002; Skaricic *et al*, 2008; Balazs *et al*, 2011). In general, however, DNA-based passive immunization suffers from the risk of genomic integration, potentially causing fatal mutations. Moreover, for safety reasons, transient vectors or regulated expression would be preferred for clinical use (Deal & Balazs, 2015). In addition, immunogenicity of virus particles has to be considered a main obstacle for the use of this type of vectors for DNA-mediated passive immunization.

In contrast, mRNA may offer an attractive alternative for passive immunization. Following initial studies in the early 1990s demonstrating that exogenous mRNA can direct protein expression *in vivo*, mRNA has emerged as a promising drug platform technology in recent years (Wolff *et al*, 1990; Jirikowski *et al*, 1992). Several studies have demonstrated the utility of mRNA as the basis of vaccines in cancer immunotherapy as well as to promote prophylactic protection from infectious diseases (Hoerr *et al*, 2000; Fotin-Mleczek *et al*, 2011; Petsch *et al*, 2012; Kubler *et al*, 2015; Kranz *et al*, 2016). Conclusive data have also been presented for mRNA as a platform for protein (replacement) therapies (Kormann *et al*, 2011; Kariko *et al*, 2012; Zangi *et al*, 2013; Thess *et al*, 2015; Balmayor *et al*, 2016). Physiological responses elicited in primates and domestic pigs treated with an mRNA-encoded hormone have suggested the feasibility of mRNA for large animal therapies (Thess *et al*, 2015). Compared to DNA-based approaches, mRNA benefits from fewer safety issues due to its non-integrative and transient nature, the latter of which contributes to better and/or easier control of protein expression, but may be disadvantageous, if prolonged bioavailability is desired or even required. In contrast to the manufacturing of many recombinant antibodies, mRNA has cost advantages, since different sequences, and as a consequence proteins, can be produced by a generic process. Thus, mRNA-based antibodies for passive immunization may be produced at competitive costs even in cases where the product is only required in emergencies such as an influenza pandemic.

In the present study, we set out to evaluate the use of mRNA for passive immunization. To this end, we focused on two indications for which passive immunization is relevant today, rabies and botulism, that can be considered prototypes for anti-pathogen and anti-toxin therapies, respectively. To demonstrate the broad applicability of mRNA technology, we further used different antibody formats. In both disease models, high levels of *in vivo* serum expression were obtained that conferred full protection in pre- and post-exposure scenarios. The flexibility of the mRNA technology was further illustrated by complementary data from additional indications, including the demonstration of therapeutic efficacy in a tumor model. Thus, our work establishes the foundation for development of novel passive immunization therapies.

# Results

## *In vitro* expression and functionality of mRNA-encoded antibodies

The intended goal of this study was to seek proof-of-principal data demonstrating the feasibility of passive immunization by means of prophylactic and therapeutic mRNA treatments encoding antibodies. Efficacious treatment with therapeutic proteins depends on both timely delivery and achieving protective levels. To demonstrate that mRNA meets these requirements, two representative and clinically relevant disease models were chosen. The first model, rabies, is an invariably fatal disease that demands rapid administration of rabies immunoglobulins in post-exposure scenarios to prevent encephalitis and finally death of infected individuals. In 2003, a cocktail of mAbs was developed which protected Syrian hamsters from lethal challenges with a highly virulent rabies virus strain in post-exposure scenarios (Prosniak *et al*, 2003). For the present study, we selected the monoclonal antibody S057, also well known as CR57, one component of the published mAb cocktail, because it provides broad neutralization of a variety of rabies virus strains via binding to glycoprotein G. The second model, botulism, is a rare but often fatal toxin-mediated illness that occurs following ingestion of food-borne bacterial spores or via pre-formed toxin. This latter exposure route is of concern as botulinum toxin is the most potent toxin known, is quite stable, has a history of use as a bioweapon, and is thus classified as a Category A bioterror agent. Due to the potency and rapid onset of symptoms, exposure to botulinum toxin demands immediate anti-toxin therapy which is currently an antiserum (Pediatrics, 1997). To demonstrate the flexibility of mRNA-mediated antibody expression, we selected a different antibody format, a camelid heavy-chain-only $V_H$ domain (VHH)-based neutralizing agent (VNA). In general, VNAs offer some advantages over classical antibodies such as ease to engineer, improved heat and pH stability, and better tissue penetration (Hamers-Casterman *et al*, 1993; van der Linden *et al*, 1999; Tillib *et al*, 2013). The use of parenterally administered VNAs that potently neutralize botulinum neurotoxin serotype A (BoNT/A) has been previously reported (Mukherjee *et al*, 2014). To further test the broad applicability of an mRNA-based passive immunization platform, results from the two main models were complemented by data with additional antibodies of the IgG mAb or VNA type, including an anti-tumor application.

Since the major objective of passive immunization is protection mediated by achieving high serum titers, we chose the liver as the target organ for mRNA-mediated protein expression. To provide prolonged mRNA stability and efficient protein translation, a previously approved format for liver expression harboring crucial regulatory mRNA elements was used (Fig 1A and D; Thess *et al*, 2015). For VNA expression, we resorted to a recently developed design for prolonged serum half-life (Mukherjee *et al*, 2014). Basically, VNA sequences included two fused VHHs complemented by an albumin-binding peptide to increase serum persistence and epitope tags for protein detection (Fig 1A). In order to provide maximal flexibility for the expression of IgG mAbs in the present study, we decided to encode heavy and light chain on separate mRNA molecules (Fig 1D). In B cells, solitary heavy chains are retained at the endoplasmic reticulum by the chaperone binding immunoglobulin protein (BiP) (Hendershot *et al*, 1987). Heavy-chain retention for

mRNA-mediated expression was confirmed in overexpressing non-B cells, including a hepatic cell line that may mimic *in vivo* conditions in the liver, our target organ for *in vivo* delivery (Appendix Fig S1). In contrast, light chains were also secreted in the absence of heavy chains. Since well-matched expression of heavy and light chain is critical for obtaining a maximum of functional antibody, we titrated the ratio between mRNAs encoding heavy and light chain *in vitro*. High expression was obtained over a broad range of molar ratios (heavy:light chain) (Appendix Fig S2). According to these results, a ratio of approximately 1.5:1 was subsequently used through the study.

Constructs for IgG mAbs against rabies glycoprotein G (S057), influenza B hemagglutinin (CR8033; Dreyfus *et al*, 2012), HIV gp120 (VRC01; Wu *et al*, 2010), and human CD20 (rituximab) all revealed strong *in vitro* expression upon mRNA transfection of BHK cells as measured by Western blot analysis and IgG-specific ELISA (Fig 1E and F, and Appendix Fig S3A–F). Likewise, expression of VNAs against botulinum neurotoxin serotype A (VNA-BoNTA) and Shiga toxin 2 (Stx2) from *E. coli* (O157:H7) (VNA-Stx2) was readily detected by Western blot and antigen-specific ELISA (Fig 1B and C, and Appendix Fig S3G and H). *In vitro* antibody mRNA half-life in primary mouse hepatocytes was approximately 15 h (Appendix Fig S4A). Functionality of antibodies secreted into cell supernatants after mRNA transfection was analyzed by various means. Specific antigen binding of mAbs against rabies, influenza, and human CD20 was verified by staining of antigen-expressing target cells (Fig 2C–E and Appendix Fig S5). Specific activity of mRNA-derived anti-HIV mAb was demonstrated to be comparable to the respective recombinant protein by measuring the inhibition of virus entry into reporter cells (Fig 2F and Appendix Fig S6). VNA-BoNTA functionality was confirmed by a SNAP25 cleavage inhibition assay demonstrating that supernatants containing mRNA-encoded VNA-BoNTA had neutralizing potency comparable to a recombinant VNA-BoNTA protein standard (Fig 2A). Finally, supernatants containing mRNA-encoded VNA-Stx2 proved capable to protect cell viability in the presence of Shiga toxin with potency comparable to recombinant VNA-Stx2 (Fig 2B). Taken together, mRNA proved to be an efficient means to express diverse antibodies and antibody formats *in vitro*.

## Expression of mRNA-encoded antibodies *in vivo*

Following the *in vitro* findings, it was explored whether protective and long-lasting serum antibody titers can be obtained in mice. Since affinity to its cognate antigen is crucial for the concentration at which a specific antibody is protective, required concentrations depend on the antibody of interest. Potential bottlenecks of reaching protective serum levels may include the mRNA delivery to and/or uptake into target cells, ribosome occupancy of mRNAs, and secretion of assembled antibodies into the blood stream. While a proven mRNA format was used to ensure efficient protein expression (see above), hepatocytes were chosen as target cells to secure efficient secretion. In order to maximize protein expression from liver, a potent lipid nanoparticle (LNP) formulation was applied. It has been specifically developed for delivery into the liver and has previously been shown to be very effective in that respect, thus providing robust protein expression (Pardi *et al*, 2015; Thess *et al*, 2015). In line with current clinical antibody applications, this LNP formulation is usually administered intravenously. Hence, this route was

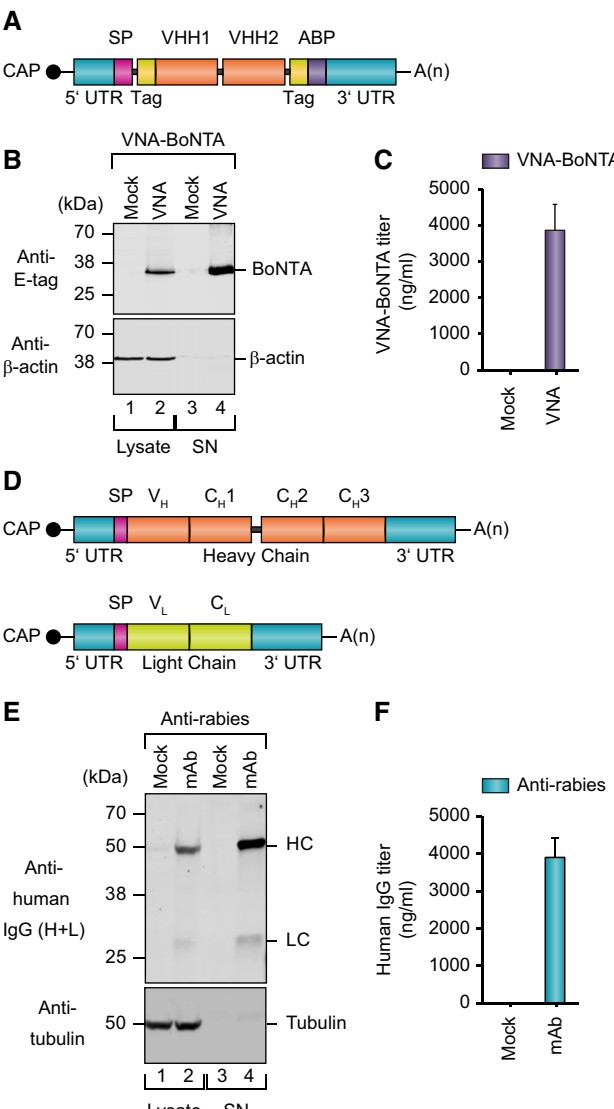

**Figure 1. *In vitro* expression of mRNA-encoded antibodies.**

A   mRNA design to encode heavy-chain-only V$_H$ domain (VHH)-based neutralizing agents (VNAs). VNAs were supplemented by affinity E-tags (Tag) and an albumin-binding peptide (ABP).

B   Western blot analysis of VNA-BoNTA expression in cell lysates and supernatants (SN) after mRNA transfection of BHK cells. Equal amounts of three replicates were pooled and loaded on denaturing SDS–PAGE. Staining for β-actin was used as loading control.

C   Quantification of VNA levels by antigen-specific ELISA of supernatants from transfected BHK cells. Titers were determined in triplicate. Supernatants from cells transfected with an irrelevant VNA were used as mock control.

D   mRNA design to encode heavy- and light-chain molecules.

E   Western blot analysis showing expression of heavy and light chains in cell lysates and accumulation in supernatants after mRNA transfection of BHK cells. Equal amounts of three replicates were pooled and loaded on denaturing SDS–PAGE. Signals correspond to heavy chain (HC), light chain (LC) and, as loading control, tubulin.

F   Quantification of mAb levels in supernatants of BHK cells by IgG-specific ELISA. Titers were determined in triplicate.

Data information: For mock transfection, an mRNA encoding eGFP was used. UTR, untranslated region; SP, signal peptide; A(n), poly(A)-tail. Results in (C and F) are expressed as means ± SD.
Source data are available online for this figure.

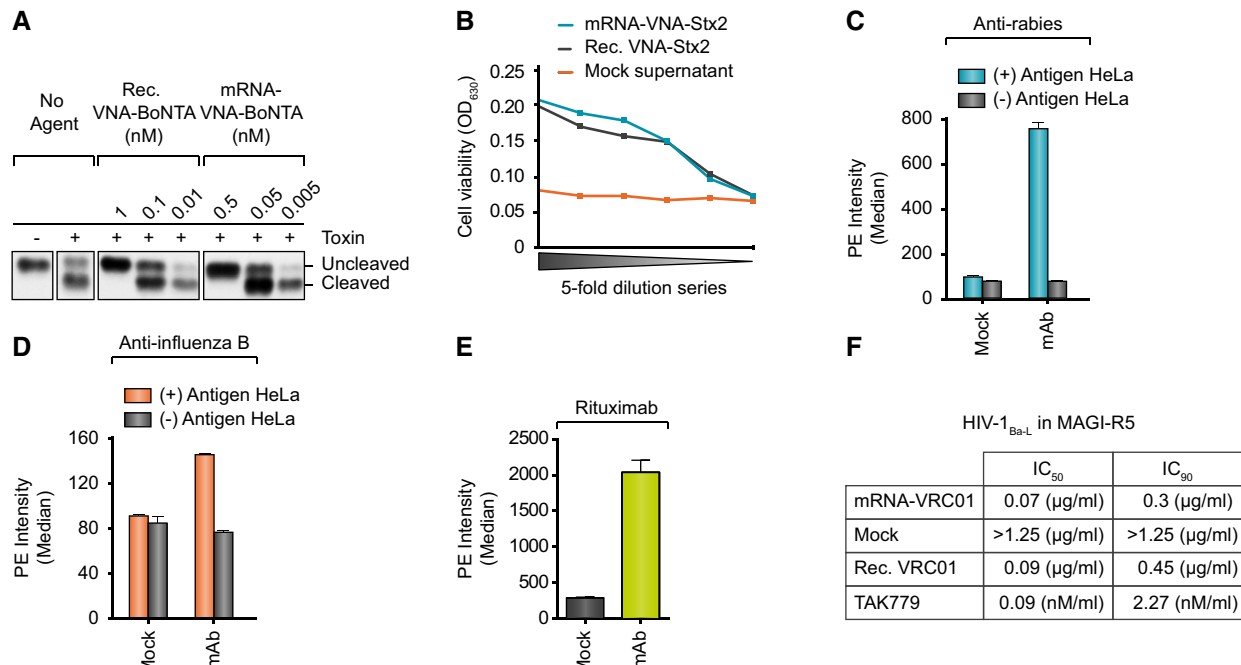

**Figure 2.  Evaluation of antibody function *in vitro*.**

A    BoNT/A toxin was incubated either with different doses of recombinant VNA, spiked into supernatant of mock-transfected BHK cells, or different dilutions of supernatant of BHK cells transfected with mRNA encoding VNA-BoNTA before SNAP25 cleavage assay. In this assay, BoNT/A cleaves SNAP25 which can be inhibited by VNA binding to the toxin. Samples from cleavage assay were loaded on denaturing SDS–PAGE and BoNT/A-mediated cleavage of endogenous SNAP25 was analyzed.

B    Cell-based toxicity assay to demonstrate the function of RNA-encoded VNA-Stx2. Supernatants from BHK cells transfected with VNA-encoding mRNA were analyzed in a dilutions series (11 nM initial concentration). Serial dilutions of supernatants from mock-transfected cells served as negative control. Recombinant VNA-Stx2 was spiked into mock-transfected medium at an initial concentration of 10 nM.

C, D   Binding of mRNA-encoded anti-rabies mAb (C) or anti-influenza B mAb (D) expressed in BHK cell supernatants to antigen-positive or antigen-negative HeLa cells. Depicted is the median of phycoerythrin (PE) fluorescence of all living cells. Fluorescence was measured in triplicate. For mock transfections, supernatants of eGFP-mRNA transfected cells were used.

E    Binding of mRNA-encoded rituximab expressed in BHK cells to Raji cells. Depicted is the median of phycoerythrin (PE) fluorescence of all living cells. Fluorescence was measured in triplicate. For mock transfections, supernatants of eGFP-mRNA transfected cells were used.

F    Supernatants of BHK cells transfected with mRNA encoding VRC01 or untransfected BHK cells were subjected to a Magi R5-Tropic Antiviral Assay. Depicted are the mAb concentrations that produced either 50% ($IC_{50}$) or 90% inhibition ($IC_{90}$) of virus entry. Recombinant VRC01 mAb or TAK779 inhibitor was used as positive controls. Individual inhibition curves are shown in Appendix Fig S5.

Data information: Results in (C, D and E) are expressed as median ± SD.

used throughout the present study, although for instance, the development of a therapy for rabies will most likely require intramuscular injection.

First, the dose–response relationship upon administration of mRNA-LNP was determined for two humanized mAbs by analyzing mouse sera with an IgG-specific ELISA. A single injection of increasing mRNA-LNP doses induced rising serum levels of antibody without reaching peak levels up to the highest doses tested in this study (Fig 3A and Appendix Fig S7A). The highest levels observed were in the low two-digit μg/ml range for both antibodies. For the anti-rabies mAb, the dose–response relationship was confirmed by analyzing virus neutralization titers which were comparable to those from an earlier active immunization study and above the protective level of 0.5 IU/ml as defined for humans at all mRNA doses tested but the lowest (Fig 3B; Schnee *et al*, 2016). Hence, as for *in vitro* expression, mRNA-mediated mAb expression *in vivo* efficiently produced functional antibodies. While maximum expression levels observed for VNA-Stx2 were almost the same as for the

mAbs according to an antigen-specific ELISA of sera, VNA-BoNTA gave rise to about 20-fold higher protein serum titers (Fig 3E and Appendix Fig S7B–D). We hypothesize that this difference can be attributed to the fact that we did not apply individually optimized mRNAs but a fixed mRNA design. Interestingly, there appears to be more than a strictly linear increase of serum titers with mRNA doses for mAbs. However, this effect is not related to the necessity of correctly assembling heavy and light chains and the separation of both on distinct mRNA molecules, since the nonlinearity is equal or even more pronounced for the single-chain VNA constructs.

Next, we addressed the kinetics of protein availability as a further important parameter for any antibody therapy. In this context, two fundamental questions are of importance: How quickly are therapeutically effective serum titers reached and how long do they persist? The ramp-up time of antibody serum titers is determined by the kinetics of delivery and uptake of mRNA-LNP into hepatocytes, protein translation, and secretion. Serum half-life of antibodies is affected by both mRNA and protein half-life. In

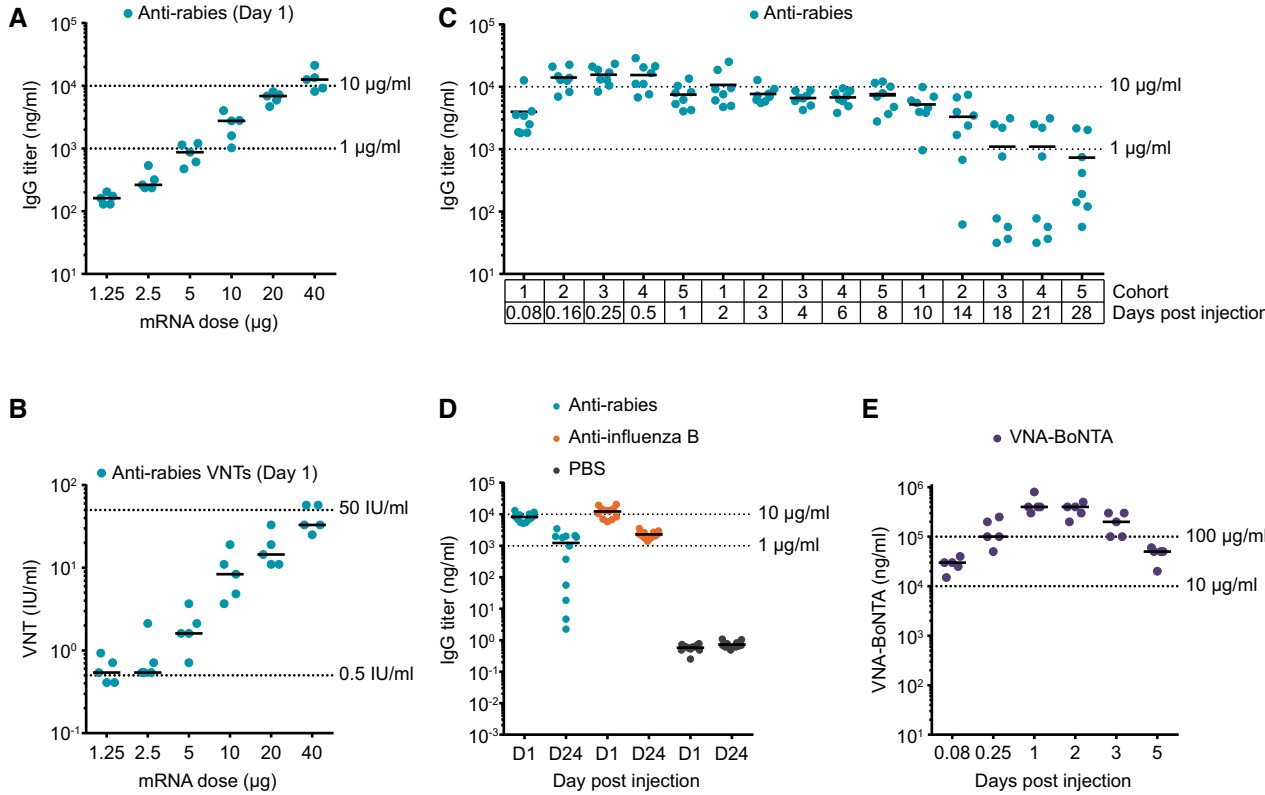

**Figure 3.  *In vivo* expression characteristics of antibody-encoding mRNA.**

A   Quantification of mAb titers in mice by IgG-specific ELISA of sera obtained 24 h after a single intravenous injection of increasing amounts of mRNA-LNP encoding anti-rabies mAb. Each group consisted of five animals.

B   Same sera as in (A) were subjected to a fluorescent antibody virus neutralization (FAVN) assay to determine anti-rabies virus neutralizing titers (VNTs in international units).

C   Quantification of mAb titers by IgG-specific ELISA at various times after a single intravenous injection of 40 µg of mRNA-LNP encoding anti-rabies mAb. Five cohorts of eight mice each were alternately bled to enable the given sampling schedule.

D   Quantification of anti-rabies and anti-influenza B mAb titers by IgG-specific ELISA 1 or 24 days post-injection of 40 µg mRNA-LNP. Each group comprised 12 mice.

E   Quantification of VNA-BoNTA titers by antigen-specific ELISA of sera at various times after a single injection of 40 µg of mRNA-LNP encoding VNA-BoNTA. For each sampling, a separate cohort of mice, each comprising five animals, was used.

Data information: Individual measurements as well as means are given for each dose and time, respectively.

the case of mAbs, the latter is primarily determined by their Fc region. In the absence of such a domain, the half-life of the VNAs used here is substantially extended by the attached albumin-binding peptide (Dennis *et al*, 2002) and have a half-life of about 1 day in mice (Mukherjee *et al*, 2014), significantly shorter than the half-life typical for mAbs. For the anti-rabies mAb, serum titers were readily detectable 2 h after treatment and peaked 6–12 h post-injection (Fig 3C). During the following days, titers slowly decreased with a half-life of approximately 1 week. While this half-life remained unchanged in about 50% of the animals during the observation period of 4 weeks, the remaining mice revealed an accelerated drop of antibody serum titers from day 10 after treatment on. This bipartite kinetics was confirmed in a second independent study. While about half of the animals showed serum half-lives of approximately 1 week for the 24-day observation period, the others were characterized by substantially lower serum titers at the end of the study (Fig 3D). In contrast, mice that received mRNA coding for anti-influenza mAb all revealed the same high serum levels after 24 days (Fig 3D). The estimated

serum half-life of this antibody was also about 1 week. For VNA-BoNTA and VNA-Stx2, considerable titers were measured as early as 2 h post-injection (Fig 3E and Appendix Fig S7B). Peak serum levels were reached 6–24 h after administration. Thus, early kinetics appeared similar for mAbs and VNAs. However, serum titers dropped much faster for VNAs than for mAbs after day 2. For both VNAs, serum half-lives were calculated to be in the range of 24–36 h which is in line with a previous report (Mukherjee *et al*, 2014). To arrive at an estimation for mRNA half-life *in vivo* in order to assess its contribution to protein bioavailability, we utilized mRNA-LNPs encoding the short-lived hormone erythropoietin, suggesting a half-life of roughly 15 h (Appendix Fig S4B), which is in good accordance with *in vitro* measurements.

To get insights into potential reasons for the accelerated serum clearance of anti-rabies antibody in some animals, we investigated the emergence of ADA (anti-drug antibody) responses. Obviously, the individual clearance rates strongly correlated with the development of a humoral response against the anti-rabies antibody (Appendix Fig S8), thus perfectly explaining the divergent kinetics.

In addition, mice were analyzed for general tolerability of mRNA-LNP treatment. We did not observe any adverse events throughout the present studies. There was a transient weak increase of some cytokines in circulation (Appendix Table S1), which obviously did not hamper high protein expression. Of note, equal or even higher levels were considered unproblematic in a recent study on modified mRNA encoding factor IX (Ramaswamy *et al*, 2017). Histopathology of liver, the target organ of mRNA-LNPs, did not reveal any signs of abnormality or inflammation (Appendix Fig S9). In summary, we could demonstrate that mRNA-LNP enables a rapid and strong expression of mRNA-encoded mAbs and VNAs. Moreover, sustained protein availability was not compromised by *in situ* expression via mRNA.

**Prophylaxis and therapy with mRNA-encoded antibodies in mice**

Based on the promising *in vivo* expression levels and kinetics, we then asked whether mRNA-mediated antibody expression may elicit measurable therapeutic effects. To this end, various biological models were used for challenge studies. Initially, mice received a single dose of mRNA-LNP encoding anti-rabies mAb and were challenged 24 h later by intramuscular (IM) virus injection (Fig 4A). Since this route resembles the most common route of rabies infection, IM challenge was used throughout the present study. All animals receiving mRNA-LNP encoding anti-rabies mAb were protected, whereas all control animals receiving mRNA-LNP encoding the negative control anti-influenza mAb succumbed

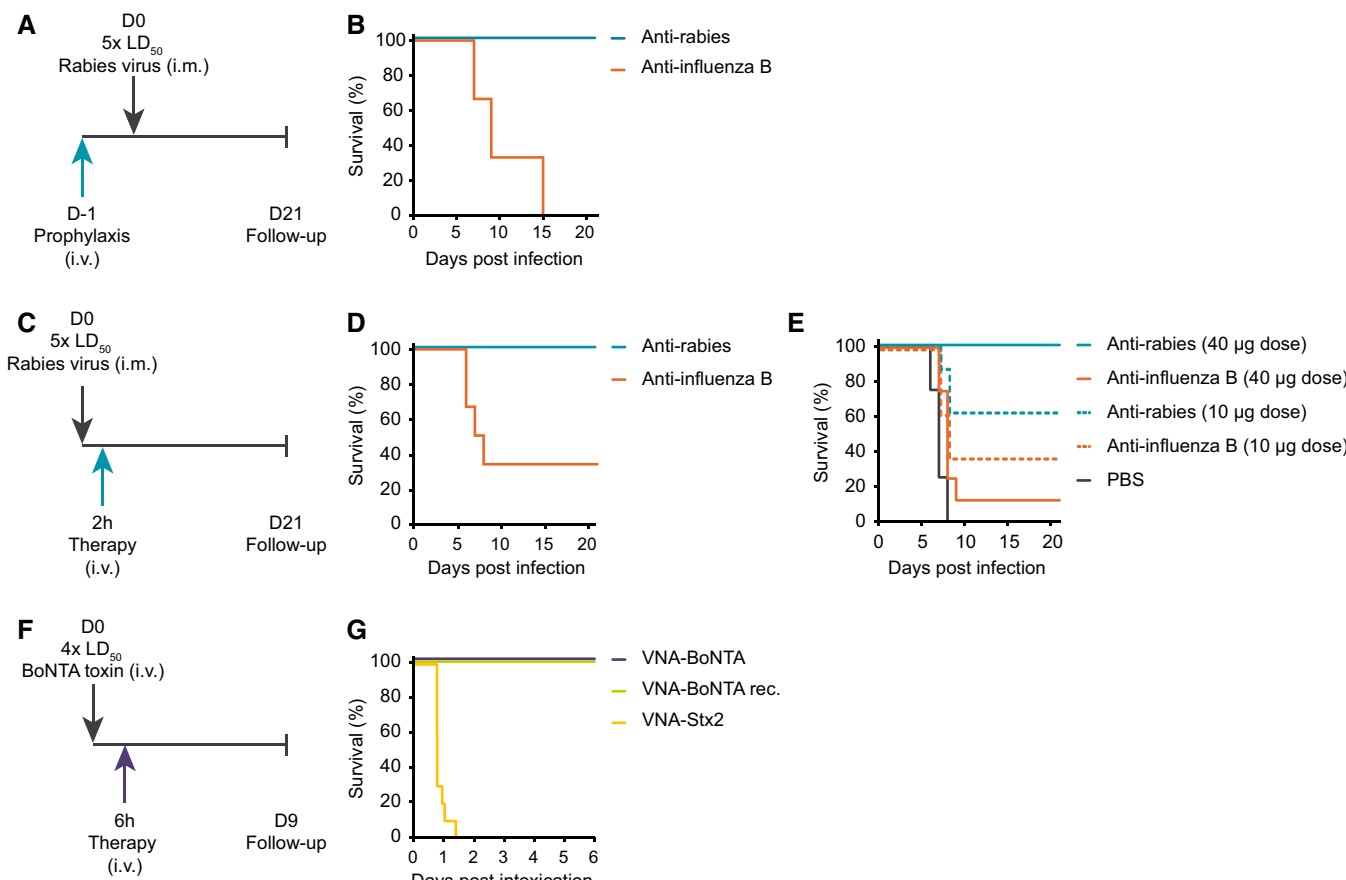

**Figure 4.  mRNA-encoded mAbs provide both prophylactic and therapeutic protection against viral or toxin challenges.**

A   Schedule of prophylactic treatment before rabies challenge. Mice received a single intravenous (i.v.) injection of 40 μg of mRNA-LNP either encoding anti-rabies or anti-influenza B mAbs 1 day before a lethal 5 × LD$_{50}$ intramuscular (i.m.) rabies challenge. Body weight, clinical score (for both see Appendix Table S2), and survival were monitored for up to 21 days.

B   Survival curve of mice that were treated according to the schedule in (A). A total of six mice per group were challenged.

C   Schedule of therapeutic treatment after rabies challenge. Mice received a single intravenous injection of 40 μg of mRNA-LNP either encoding anti-rabies or anti-influenza B mAbs 2 h after a lethal 5 × LD$_{50}$ intramuscular rabies challenge. Body weight, clinical score (for both see Appendix Tables S3 and S4), and survival were monitored for up to 21 days.

D   Survival curve of mice treated according to the schedule in (C). A total of six mice per group were challenged.

E   Survival curve of mice that were treated according to schedule (C) but received different doses of mRNA-LNP as well as buffer as further control. A total of eight mice per group were challenged.

F   Schedule of therapeutic BoNT/A challenge. Mice received a single intravenous injection of 2 μg of recombinant VNA-BoNTA or 40 μg of mRNA-LNP either encoding VNA-BoNTA or VNA-Stx2 6 h after a lethal 4 × LD$_{50}$ intravenous BoNT/A challenge. Survival was monitored for up to 6 days.

G   Survival curve of mice that were treated according to the schedule in (F). A total of ten mice per group were challenged.

(Fig 4B). Next, we switched to a clinically more relevant scenario, post-exposure prophylaxis. After intramuscular infection, rabies virus travels along the peripheral afferent nerves in order to reach the central nervous system where it rapidly causes lethal encephalitis. The time window for an efficacious treatment is limited by the time it takes until the virus finally reaches the brain. In mice, rabies infection is believed to proceed rapidly. Thus, in order to be most probably within the yet undefined time window, mRNA-LNP was administered 2 h after infection (Fig 4C). All mice expressing the anti-rabies mAb survived, whereas the majority of control animals which received anti-influenza mAb mRNA succumbed (Fig 4D).

To clarify whether the partial survival of control animals represents a statistical outlier or may be indicative of some form of non-specific immunostimulation, a second post-exposure challenge experiment with additional control groups and larger group size was conducted. Again, all mice receiving 40 μg of mRNA for anti-rabies mAb survived (Fig 4E). A fourfold lower dose gave rise to just partial protection. On the one hand, this means that efficacy of this specific antibody is limited, at least requiring titers in the upper single digit μg/ml range for full protection. On the other hand, this clearly indicates that the VNT guidelines for humans cannot be simply transferred to mice, since 10 μg of mRNA gave VNTs well above the 0.5 IU/ml threshold (Fig 3B). While all animals receiving just buffer succumbed to rabies infection, 1 out of 12 mice treated with 40 μg of mRNA for the irrelevant anti-influenza mAb survived, which is well below the survival rate in the first experiment. However, animals that received a fourfold lower dose of influenza mRNA again showed a survival rate of about 40%. In case of a non-specific protection by mRNA-LNP, survival rates would be expected to be the opposite. Hence, survival rates of 0–40% rather appear to represent the natural variation of spontaneous survival under the challenge conditions used here.

To further challenge the kinetics of protection by mRNA-encoded antibodies, we next switched to a model in which immediate action is crucial for survival, BoNT/A intoxication. BoNT/A challenge experiments are well established (Mukherjee *et al*, 2012) and represent an excellent system for analysis, since the time course of intoxication is well defined and unprotected mice succumb rapidly. Mice were challenged with $4 \times LD_{50}$ of BoNT/A intravenously and treated with mRNA-LNP either encoding VNA-BoNTA or VNA-Stx2 or with recombinant VNA-BoNTA at 2, 4 or 6 h post-intoxication (Fig 4F and Appendix Fig S10).

Full survival with no clinical signs was observed when animals were treated at 2 or 4 h after intoxication with either VNA-BoNTA mRNA or recombinant VNA-BoNTA (Appendix Fig S10A and B), while all control mice died. Mice treated 6 h post-intoxication with VNA-BoNTA mRNA or recombinant VNA-BoNTA also survived (Fig 4G); however, these mice displayed moderate clinical signs. The presence of clinical signs indicates that 6 h is near the end of the treatment window, since antitoxins cannot neutralize toxin once it has entered and paralyzed motor neurons. Again, all mice which received the negative control, mRNA-LNP encoding VNA-Stx2, succumbed within 36 h. Taken together, mRNA was at no disadvantage compared to recombinant protein in a post-intoxication setting, thereby highlighting the highly favorable kinetics of mRNA-based antibody expression onset which opens up a broad array of new therapeutic approaches.

We finally set out to evaluate whether mRNA-mediated antibody expression may be effective in the field of cancer immunotherapy where mAbs are widely used in medical practice. For instance, the combination of chemotherapy and intravenous administration of rituximab is the gold standard for treating non-Hodgkin's lymphoma. Hence, we inoculated mice intravenously with luciferase expressing Raji lymphoma cells and started treatment with mRNA-LNP encoding rituximab 4 days later (Fig 5A). Control animals all revealed strong tumor cell proliferation and had to be euthanized 17 days after inoculation due to severe symptoms (Fig 5B–D). By contrast, repeated intravenous administration of either 50 or 10 μg of mRNA-LNP for rituximab strongly decelerated or even abolished tumor cell growth. Until the end of the study 28 days after tumor cell inoculation, all animals in the high-dose group and all but two mice in the low-dose group survived. In a second study, rituximab-encoding mRNA-LNPs were compared to recombinant protein (Fig 5E and F). Moreover, mRNA-LNPs coding for an irrelevant antibody were used as further control. As in the previous study, untreated control mice revealed fast tumor progression leading to euthanization after about 17 days. Animals treated with irrelevant antibody mRNA appeared to show delayed tumor growth kinetics. This effect may be attributable to the aforementioned weak cytokine response and/or repeated injections/handling per se. Nevertheless, even compared to this control group, treatment with recombinant antibody at a dose which has been widely used in mouse studies on rituximab clearly suppressed tumor growth. The anti-tumor effect of 50 μg of mRNA-LNP for rituximab was even more pronounced and as strong as in the first study. In summary, mRNA-encoded antibodies appear to represent a viable therapeutic option for various biological threats, including (viral) infections, intoxication, and cancer.

## Discussion

For any viable platform for prophylactic and therapeutic antibodies, three parameters are of particular relevance: antibody serum titers sufficient to fully protect against biological threats, a fast ramp-up of serum titers upon treatment, and an acceptable half-life of serum titers, mainly for the sake of convenience in terms of less frequent treatments. In the present work, we set out to evaluate whether mRNA meets these requirements and therefore represents a competitive antibody platform. Using diverse disease models, mRNA-mediated antibody expression proved capable of providing therapeutic benefit. It conferred full protection against intoxication and virus challenge and could eradicate neoplastic cells in a murine tumor model. While the present manuscript was under review, the feasibility of using mRNA for passive vaccination was independently confirmed in an HIV model (Pardi *et al*, 2017).

An earlier study on a different secreted protein, erythropoietin, revealed that findings in a murine model translated to larger animals such as domestic pigs as well as to primates (Thess *et al*, 2015). Thus, the present data suggest that mRNA-encoded antibodies offer a therapeutic option in humans as well. However, therapeutic doses of today's recombinant antibodies are often quite high. Are even those indications accessible to mRNA approaches? Several considerations argue in favor of such a conclusion. Firstly, it is possible that *in situ* antibody expression may reduce the amount of protein

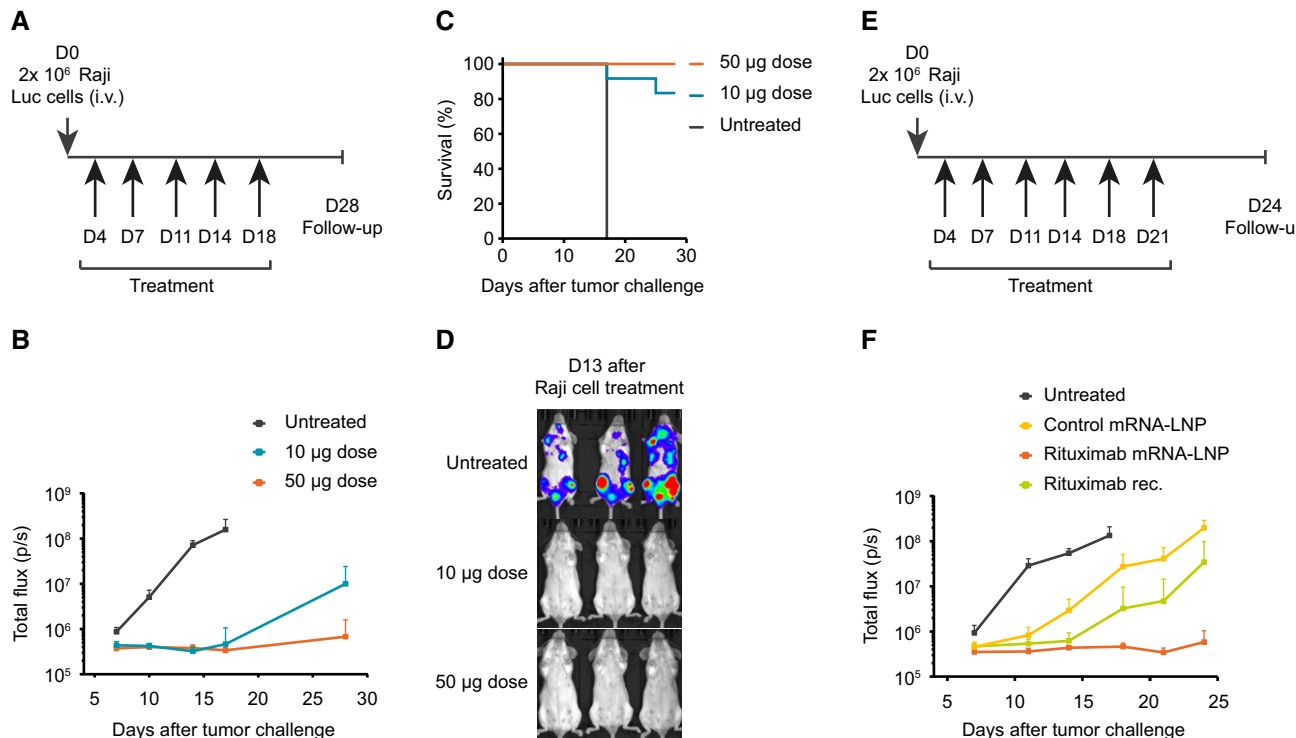

**Figure 5.  mRNA-encoded mAb protects mice from lethal tumor challenge.**

A   Schedule of tumor challenge and mRNA treatments. On Day 0, mice received an intravenous (i.v.) administration of $2 \times 10^6$ Raji Luc cells. At the indicated times (lower arrows), mice received intravenous injections of either 10 or 50 μg of mRNA-LNP encoding rituximab. Survival and tumor development were monitored for up to 28 days.

B   Tumor development was assessed by whole-body luminescence imaging at the indicated times after tumor challenge. Each group comprised 12 mice.

C   Survival of mice treated according to the schedule in (A). Each group comprised 12 mice.

D   Representative luminescence images of mice either treated with two different doses of mRNA-LNP encoding rituximab or untreated mice at day 13 after tumor challenge.

E   Schedule of tumor challenge and mRNA treatments. On Day 0, mice received an intravenous (i.v.) administration of $2 \times 10^6$ Raji Luc cells. At the indicated times (lower arrows), mice received intravenous injections of 50 μg of mRNA-LNP encoding rituximab or control antibody or 200 μg of recombinant rituximab. Survival and tumor development were monitored for up to 24 days.

F   Tumor development of experiment as in (E) was assessed by whole-body luminescence imaging at the indicated times after tumor challenge. Each group comprised 12 mice.

Data information: Results in (B and F) are expressed as means ± SD.

required for a therapeutic effect. Secondly, up to the doses tested so far, we found that the dose–response curve for mRNA-mediated antibody expression did not reveal any saturation nor did we observe any dose-limiting toxicity. Thus, a further dose escalation appears to be feasible if even higher serum antibody levels are desirable. Thirdly, in light of previous findings, it is likely that target-specific mRNA optimizations and further improvements to the formulation can substantially increase efficacy (Thess *et al*, 2015). This assumption is corroborated by the differences in peak expression levels between anti-rabies mAb and BoNTA VNA in the present study (Fig 3A and E). Obviously, this difference cannot be simply attributed to the distinct antibody designs, since Stx2 VNA expression produced antibody titers at the same level as the anti-rabies mAb. Thus, the use of a previously approved design for the various constructs instead of having individually optimized mRNA formats is most likely responsible for the different outcomes. Fourthly, the effective dose of an antibody is strongly dependent on its characteristics, that is, potential limitations may be overcome with the right protein (De Benedictis *et al*, 2016). Overall, with

regard to antibody titers, mRNA appears to be applicable to a broad range of indications. However, this will certainly have to be confirmed on a case-by-case basis. Moreover, indications such as rabies will probably require intramuscular administration in the light of approved therapies. It has still to be demonstrated that this route enables protective expression levels, particularly since it may reduce the number of target cells reached by the mRNA. Of note, mRNA appears to be at no (substantial) disadvantage compared to DNA with respect to serum levels.

Any passive vaccination platform needs to provide effective antibody levels soon after drug administration. We now demonstrate herein that our mRNA technology enables the *in vivo* synthesis of antibodies displaying favorable pharmacokinetics with substantial antibody titers already detectable as early as 2 h after treatment. To address whether this temporal profile translates into a fast onset of protection, infection with rabies virus is a suitable *in vivo* model. According to *in vitro* data, the virus enters cells within 20 min after inoculation (Dietzschold *et al*, 1985). Nevertheless, antibodies could inhibit infection up to 4 h post-exposure. *In vivo*, many anti-rabies

antibodies become ineffective after the virus has entered the CNS. Whether this applies to the S057 mAb used here is unknown. Furthermore, the time window for efficacious antibody intervention has not been defined for mice for which a very fast progress of the disease has been reported. Hence, we chose a 2 h post-exposure scenario for the mRNA and could demonstrate full protection. To further challenge the kinetics of protection, we drew on an intoxication model that requires even faster responses. Protection against death was achieved even with a 6 h post-exposure treatment. Thus, although mRNA clearly cannot act as fast as an infused recombinant antibody, our data suggest that mRNA may meet the requirements of at least most indications, if not all.

For the present study, LNP-formulated mRNA was used to express humanized IgG mAbs in mice. Serum half-life for two different proteins turned out to be in the range of 1 week. This temporal profile is primarily determined by the half-life of the antibody, since longevity of IgG proteins substantially exceeds that of mRNA. In general, in case of long-lived proteins, the use of an mRNA expression platform has no apparent impact on the duration of the therapeutic effect, but mRNA half-life does contribute to peak level expression. By contrast, short-lived proteins can significantly profit by being expressed from mRNA (Kariko *et al*, 2012; Thess *et al*, 2015). Accordingly, VNAs with an estimated serum half-life of 1–2 days (with albumin-binding) should benefit from being encoded by mRNA (Mukherjee *et al*, 2014). Indeed, estimated half-lives of VNA serum titers at one to 3 days after treatment were on average 1.5-fold higher than from day 3 on, even without a target-specific mRNA optimization. While during the first phase mRNA and protein half-lives both contribute to the kinetics of serum titers, the kinetics during the second phase is almost exclusively determined by the protein's properties. In contrast, DNA has the potential to elicit longer bioavailability which may be of advantage in pandemic scenarios, but may be considered disadvantageous, if persistent antibody expression is neither required nor desired.

Any protein therapy has the inherent risk of inducing an anti-drug antibody (ADA) response which may limit drug efficacy, particularly over time. The probability of an ADA response is especially high for proteins that are foreign to the body. The risk may be further increased, if the protein is encoded by a nucleic acid which may trigger an innate immune response via diverse cellular sensors. Hence, to test for induction of ADA using an mRNA-mediated antibody therapy, we chose humanized mAbs for expression in mice. The pharmacokinetics data for an anti-rabies antibody indeed appeared to suggest an induction of an ADA response as about half of the animals revealed accelerated serum clearance after about 2 weeks (Fig 3C and D). This drop of serum titers correlated with elevated ADA levels 4 weeks after treatment (Appendix Fig S8). In contrast, serum clearance maintained its slow initial rate during the whole observation period for a humanized anti-HA antibody (Fig 3D). This clearly suggests that our chemically unmodified mRNA does not stimulate a suppressive immune response per se which is in line with a previous report on erythropoietin (Thess *et al*, 2015). Instead, the ADA response is triggered by the mismatch between antibody sequence and species and strongly dependent on the protein's identity. Thus, mRNA appears to represent an appropriate platform for antibody expression. As expected, half-lives of serum titers were much

shorter for VNAs compared to mAbs. However, better protein stabilization approaches and further improved mRNA half-life by specific mRNA optimization will certainly reduce the gap between VNAs and mAbs for therapies in which persistence in serum is preferred.

Here, we present evidence that mRNA can act as an appropriate platform for passive immunization. In addition, it has been previously demonstrated that mRNA represents a potent vaccine technology (Kreiter *et al*, 2010; Fotin-Mleczek *et al*, 2011). In consequence, mRNA may be the ideal platform for applications where combination of both passive and active immunization is of advantage or even required such as for rabies post-exposure prophylaxis. Ideally, passive and active immunization will use the same route of administration for the sake of convenience. It has been demonstrated that mRNA active vaccination can use both the intramuscular as well as the intravenous route (Petsch *et al*, 2012; Kranz *et al*, 2016). Moreover, expression of therapeutic proteins can also be achieved by intramuscular administration (Schlake *et al*, 2012). Therefore, a merging of administration routes for passive and active immunization appears to be possible, if necessary or desirable. Taken together, the findings of the present study suggest mRNA to be a promising novel platform for antibody therapies.

## Materials and Methods

### Protein design

Antibody sequences were designed as follows. The sequences of anti-rabies antibody SO57 heavy (GenBank: AAO17821.1) and light chain (GenBank: AAO17824.1) have been published previously (Prosniak *et al*, 2003). The sequences of the variable regions of CR8033 heavy (PDB: 4FQL_H) and light chain (PDB: 4FQL_L) were published by Dreyfus *et al* (2012). The sequences for variable regions of VRC01 have been described by Wu *et al* (2010). The sequence of rituximab variable regions has been published previously (Nebija *et al*, 2011). For the heavy chains of CR8033 and VRC01, the sequence of the IgG constant region of SO57 was used. For rituximab, the immunoglobulin heavy chain was derived from AFR78282.1. For CR8033, VRC01, and rituximab, the sequence of an immunoglobulin kappa light-chain constant region was used (GenBank: AGH70219.1). VNA sequences for both VNA-BoNTA and VNA-Stx2 consist of two VHHs each, and their sequences have been published. VNA-BoNTA is reported in Mukherjee *et al* (2014). VNA-Stx2 has the same sequence as VNA-BoNTA except that the two BoNT/A-neutralizing VHHs were replaced by two Stx2-neutralizing VHHs, Stx-A5, and Stx2-G1, which were reported in Tremblay *et al* (2013). MfEPO (*Macaca fascicularis*) is deposited as P07865 (Uniprot).

### mRNA sequences and synthesis

The design and synthesis of mRNA sequences have been described previously (Thess *et al*, 2015). In brief, mRNAs for MfEPO or encoding anti-rabies (SO57) or anti-influenza B (CR8033) mAb as well as VNA sequences contained a 5′ cap structure, 5′ UTR (Thess *et al*, 2015), open reading frame, 3′ UTR (Thess *et al*, 2015), and poly-A sequence followed by a C30 stretch and a histone stem loop

sequence. The mRNA sequences encoding VRC01 and rituximab contained a 5′ cap structure, open reading frame, 3′ UTR (CRE of alpha globin), and a poly-A sequence followed by a C30 stretch. Sequences were codon-optimized for human use and did not include chemically modified bases. While all mAb-mRNAs were co-transcriptionally capped, mRNAs encoding VNAs or MfEPO were capped enzymatically using ScriptCap 2′-O-methyltransferase (Biozym, Cat. 150360) and ScriptCap m7G Capping System (Biozym, Cat. 150355). For *in vivo* studies, mRNAs were further polyadenylated using A-Plus Poly(A) Polymerase Tailing Kit (Biozym, Cat. 150491). Capping and polyadenylation were carried out according to manufacturer's recommendations.

## RNA formulation

Lipid nanoparticle formulation was conducted at Acuitas Therapeutics (Vancouver, Canada) and has been described previously (Thess *et al*, 2015). RNAs encoding mAbs were mixed at a molar ratio of approximately 1.5 (heavy- over light-chain-encoding mRNA) before LNP formulation. For injections, mRNA-LNP was diluted in phosphate-buffered saline pH 7.4.

## Cell transfections

For *in vitro* transfection of cells, RNAs were complexed with Lipofectamine 2000 (Life Technologies, Darmstadt, Germany; at 1.5 μl/μg of mRNA for BHK or at 1 μl/μg for HepG2) and transfected into cells according to manufacturer's instructions. To analyze the expression of different mAbs and VNAs, 400,000 BHK cells were seeded 1 day before transfection in six-well plates. Cells were transfected with 5 μg of antibody or VNA-encoding RNAs. Approximately 2 h after transfection, transfection mix was replaced by 1.5 ml of serum-free freestyle 293 medium (Thermo Scientific Cat. 12338018). Cells were grown for approximately 48 h, and supernatants were harvested, spun down to remove cell debris, and used for analyses. For preparation of lysates, cells were incubated for 5 min with Laemmli buffer and subsequently collected. To analyze different heavy-to-light-chain ratios, 10,000 BHK cells were seeded 1 day before transfection in 96-well plates. Cells were transfected with 100 ng of antibody-encoding RNAs. Approximately 2 h after transfection, transfection mixes were replaced by 200 μl freestyle 293 medium (Thermo Scientific Cat. 12338018). Cells were grown for approximately 48 h, and supernatants were harvested and used for analyses. Mouse primary hepatocytes (Thermo Fisher, Cat. MSCP10) were thawed according to manufacturer's instructions. Cells were plated on collagen-coated 24 well plates (Thermo Fisher, Cat. A1142802) at a density of 200,000 cells per well in a volume of 0.5 ml in plating medium (Williams medium E; Thermo Fisher Cat. 22551022 supplemented with Primary Hepatocyte Thawing and Plating Supplements; Thermo Fisher, Cat. CM3000). After 5 h, cells were washed twice in serum-free maintenance medium (Williams medium E; Thermo Fisher Cat. 22551022 supplemented with Primary Hepatocyte Maintenance Supplements; Thermo Fisher, Cat. CM4000). Transfection of cells with 0.5 μg mRNA-LNP encoding anti-rabies mAb, diluted in 100 μl Opti-MEM, was done in maintenance medium overnight. Supernatants were collected every day, and medium was exchanged. Antibody titers were measured by IgG-specific ELISA as described above.

## Western blot analysis

For all experiments, pooled triplicates of equal amounts of lysates or supernatants were loaded. For all Western blots depicted in this study, 12% SDS Tris-glycine gels were used. Proteins were transferred to a nitrocellulose membrane (Odyssey nitrocellulose membrane 0.22 μm, Li-COR Biosciences, Cat. 926-31092) and afterward blocked in 5% skimmed milk in TBST buffer (TBS containing 0.1% Tween-20 from Sigma, Cat. P2287). Membranes were first incubated with rabbit anti-α/β-tubulin 1:1,000 (New England Biolabs Cat. 2148S) in 0.5% skimmed milk in TBST for 1 h. After three washes (10 min each) in TBST, both a secondary antibody against rabbit (goat anti-rabbit IgG (H + L) IRDye® 680RD; Li-COR Biosciences Cat. 926-68071) and an antibody to detect human antibodies (goat anti-human IgG (H + L) IRDye® 800CW; Li-COR Biosciences Cat. 926-32232) were incubated at 1:15,000 in 0.5% skimmed milk in TBST for 1 h or in case of HepG2 supernatants overnight. Immediately before band detection, all membranes were washed three times each for 10 min in TBST and stored in TBS lacking Tween-20 until analysis. For detection of VNAs, the following antibodies were used: mouse anti-ß actin antibody 1:20,000 (Abcam, Cat. ab6276) and rabbit anti-E-tag antibody 1:10,000 (Bethyl, A190-133A) as primary antibodies and goat anti-mouse IgG (H + L) IRDye® 680RD 1:10,000 (Li-COR, Cat. 926-32210) and goat anti-rabbit IgG (H + L) IRDye® 800CW 1:15,000 (Li-COR, Cat. 926-32211) were used as secondary antibodies. Incubation times were the same as described above. Protein detection and image processing were carried out in an Odyssey® CLx Imaging system and LI-COR's Image Studio version 5.2.5 according to manufacturer's recommendations.

## ELISA

Goat anti-human IgG (1 mg/ml; SouthernBiotech; Cat. 2044-01) was diluted 1:1,000 in coating buffer (15 mM $Na_2CO_3$, 15 mM $NaHCO_3$ and 0.02% $NaN_3$, pH 9.6) and used to coat Nunc MaxiSorp® flat-bottom 96-well plates (Thermo Fischer) with 100 μl for 4 h at 37°C. After coating, wells were washed three times (PBS pH 7.4 and 0.05% Tween-20) and blocked overnight in 200 μl blocking buffer (PBS, 0.05% Tween-20 and 1% BSA) at 4°C. Human IgG1 control antibody (Erbitux at 5 mg/ml; Merck, PZN 0493528) was diluted in blocking buffer to 100 ng/ml. Starting with this solution, a serial dilution was prepared for generating a standard curve. Samples were diluted appropriately in blocking buffer (PBS, 0.05% Tween-20, and 1% BSA) to allow for quantification. All further incubations were carried out at room temperature. Diluted supernatants or sera were added to the coated wells and incubated for 2 h. Solution was discarded and wells were washed three times. Detection antibody (goat anti-human IgG Biotin, Dianova; Cat. 109065088) was diluted 1:20,000 in blocking buffer, 100 μl was added to wells and incubated for 60–90 min. Solution was discarded and wells were washed three times. HRP–streptavidin (BD Pharmingen™, Cat. 554066) was diluted 1:1,000 in blocking buffer, 100 μl was added to wells and incubated for 30 min. HRP solution was discarded and wells were washed four times. 100 μl of Tetramethylbenzidine (TMB, Thermo Scientific, Cat. 34028) substrate was added and reaction was stopped by using 100 μl of 20% sulfuric acid. MfEPO levels were quantified with Human Erythropoietin Quantikine IVD ELISA Kit

(R&D Systems, Cat. DEP00) according to manufacturer's instructions. For the detection of mouse-specific anti-SO57 antibodies, an ELISA was carried out as described above with the following antibodies: BHK cells were transfected with mRNA encoding SO57 as described above. After 72 h, supernatants were collected and S057 antibody was purified using protein A beads from Pierce™ (Thermo Scientific Cat. 44667) according to manufacturer's instructions. Wells were then coated with 100 µl at 1 µg/ml. Mouse anti-human IgG (H + L) cross-adsorbed secondary antibody (Thermo Scientific Cat. 31135) was used as internal standard. Goat anti-mouse IgG (H + L) secondary antibody, biotin (Thermo Scientific Cat. 31800), was used at 1:50,000 for detection. Absorbance at 450 nm was measured in a plate reader (Hidex Chameleon Model: 425-156 or BertholdTech TriStar² Model: LB 942). The measurement of VNA titers in supernatants or sera has been described previously (Mukherjee *et al*, 2014; Sheoran *et al*, 2015).

### VNA functional testing

VNA-mRNAs were transfected into BHK cells as described above, and supernatants, collected at 48 (VNA-Stx2) or 72 h (VNA-BoNTA), were tested for their ability to inhibit the effects of the respective toxins in cell-based assays. BoNT/A neutralization by culture supernatants was assessed as reported in Mukherjee *et al* (2012). Briefly, serial dilutions of the BHK supernatants were incubated with primary neurons, then treated with BoNT/A (20 pM) overnight, and finally, intoxication was assessed by analysis of cleavage of endogenous SNAP-25 by Western blotting. Stx2 neutralization was performed as reported in Tremblay *et al* (2013). Briefly, serial dilutions of the BHK supernatants were incubated with Vero cells, then treated with Stx2 (35 pM) for 2 days, and finally, cell death was quantified by protein staining. Recombinant VNA proteins were used as positive controls in both assays.

### FACS analysis

For the generation of antigen-expressing target cells, 400,000 HeLa cells were seeded in six-well plates 24 h before transfection. Cells were transfected with 1 µg of mRNAs encoding either rabies virus glycoprotein or influenza B hemagglutinin or mock-transfected with Lipofectamine 2000 only. 24 h after transfection, cells were washed once in PBS, detach buffer (10 mM Tris pH 7.5, 150 mM NaCl, and 1 mM EDTA) was added, and cells were incubated at 37°C for 15 min. Cells were collected and seeded in 96-well plates. All subsequent incubation steps were carried out for 30 min at 4°C in the dark. All washing steps were done with PBS, 0.5% BSA, separated by centrifugation at 500 rcf each time for 3 min. For discrimination of live and dead cells, LIVE/DEAD™ Fixable Dead Cell Stain Kits (Thermo Scientific, Cat. L34957) was applied first. Supernatants were then discarded, and cells were washed two times. For binding of antibodies to antigen-expressing cells, diluted supernatants containing antibodies (in PBS, 0.5% BSA) were added. Afterward, cells were washed two times, followed by addition of 1:750 diluted secondary antibody (goat anti-human IgG biotinylated, Dianova, Cat. 109065088). After two washing steps, cells were incubated with a 1:500 diluted streptavidin–R-Phycoerythrin conjugate solution (Thermo Scientific Cat. S866). If necessary, cells were fixed in 1% formaldehyde for 30 min at 4°C in the dark. For FACS analysis, cells

were re-suspended in PFEA (PBS, 2% FCS, 2 mM EDTA and 0.01% sodium azide). For the detection of functional rituximab in cell supernatants, Raji cells were seeded in 96-well plates at a density of 200,000 cells per well. For anti-CD20 staining, supernatants of HEK293T cells, transfected with rituximab as described above, were added. Staining was conducted according to the protocol given above. For quantification of rituximab in supernatants, recombinant rituximab (MabThera, Roche, PZN-8709896) served as a control. FACS analysis was carried out using a BD FACSCANTO II system (BD Biosciences).

### Cytokine analysis

Cytokines were quantified using BD Cytometric Bead Array (CBA) Mouse/Rat Soluble Protein Master Buffer Kit (BD, Cat. 558266) according to manufacturer's instructions. FACS analysis was carried out using a BD FACSCANTO II system (BD Biosciences).

### Histopathological analysis of liver

BALB/c mice (Janvier Labs, Le Genest-Saint-Isle, France) were injected intravenously with mRNA-LNP. 24 h post-injection, animals were sacrificed and isolated livers were fixed in neutral buffered formalin. Tissue samples were embedded in paraffin and cut on a rotation microtome; 5-µm thin slices were put onto microscopic slides and dried overnight. After hematoxylin and eosin (H+E) staining, liver sections were inspected for histological abnormalities.

### Animal experiments

Dose-finding studies for anti-influenza B and anti-rabies mAbs, kinetics of anti-rabies mAb expression as well as rabies challenge experiments were carried out at Viroclinics Biosciences B.V. (Rotterdam and Schaijk, Netherlands) in accordance with national laws. Ethical approvals in the Netherlands for the studies are registered as AVD905002015124-WP03, WP04 and WP05, respectively. Six- to eight-week-old, specific-pathogen-free, outbred female NIH Swiss albino mice (Harlan, The Netherlands) were used. For dose finding and expression kinetics studies, mice received a single intravenous injection of LNP-formulated mRNA. To analyze the kinetics of antibody expression, a total of five cohorts were used due to limitations as to the frequency and total volume of blood drawings. Sample size was estimated using previous unpublished data in order to enable detection of twofold differences in expression and analysis of the kinetics. All animals received a single dose of 40 µg LNP-formulated mRNA at the beginning of the study. All samplings were carried out by retro-orbital bleeding. To analyze baseline human IgG titers, naive mice were sampled either two (dose finding) or five (kinetics) days before treatment with mRNA-LNP. In all experiments, baseline human IgG titers were below the detection limit. All animal experiments involving mRNA-encoded VNAs were conducted at the Department of Infectious Disease and Global Health, Tufts Cummings School of Veterinary Medicine (North Grafton, USA) in conformance with Tufts University IACUC Protocol #G2016-74 (approval date July 18th, 2016). Six- to eight-week-old female CD1 mice (Charles River Labs, Wilmington, USA) were randomized based on body weight and received single intravenous injections of

LNP-formulated mRNA into the tail vein. Blood was sampled by retro-orbital bleeding at defined times, and VNA accumulation in sera was measured as described previously (Tremblay *et al*, 2013; Mukherjee *et al*, 2014). Sample size was estimated using previous unpublished data in order to enable analysis of the kinetics. For analysis of MfEPO expression kinetics, *in vivo* experiments were carried out with ethical approvals registered as CUR7-13. Briefly, 6- to 10-week-old female BALB/c mice were intravenously (tail vein) injected with 30 μg of mRNA-LNP encoding MfEPO with one cohort for each time point. Samplings were done by retro-orbital bleeding. In all studies, animals were housed under standard conditions with a standard commercial rodent diet and tap water provided to the animals *ad libitum*.

## Challenge experiments

Six- to eight-week-old female NIH Swiss albino mice were housed in custom-made glovebox isolators under standard conditions and were challenged with a $5 \times LD_{50}$ dose of intramuscularly applied Pasteur strain rabies virus. Sample size was estimated based on prior virus stock titration and considered the likelihood of spontaneous survivors. For both prophylactic and therapeutic treatment, a single intravenous injection of LNP-formulated mRNA (or PBS when indicated) was administered either 24 h before or 2 h after challenge. Mice were inspected daily for clinical symptoms. Mice which suffered from severe progression to rabies encephalitis were euthanized at the humane endpoint (a score of 3 or a body weight loss of more than 5% on two following days). The scoring of clinical symptoms involved: score of 0 (no symptoms), score of 1 (early sign of encephalitis), score of 2 (progression to rabies encephalitis) and score of 3 (rabies encephalitis = humane endpoint). Scoring was done by two individual operators without any information on treatment. A detailed definition of the scores can be found in Appendix Table S2. Challenge of mice with $4 \times LD_{50}$ of botulinum toxin A1 was carried out as described previously (Mukherjee *et al*, 2014). Animals were housed in standard isolator cages and randomized based on body weight. Therapeutic treatments following lethal challenges involved intravenous injections of 40 μg LNP-formulated mRNA or 2 μg of recombinant VNA-BoNTA. Sample size was estimated based on previous unpublished data on toxin challenge and mRNA-encoded antibody expression. Tumor challenge experiments were conducted at Vivopharm (Hummelstown, USA). Ethical approvals are registered as 2011-031 at Pennsylvania State College of Medicine Institutional Animal Care and Use Committee. Sample size was estimated on previous experience on the efficacy of tumor cell engraftment. A total of $2 \times 10^6$ human Raji-luc2 B-cell lymphoma cells (Raji human Burkitt's lymphoma cells, sourced from American Type Culture Collection, Rockville, MD, USA) were engrafted intravenously by tail vein injection in ten- to 12-week-old female NOD/SCID mice. Four days after tumor challenge, mice were treated with intravenous injections of either ten or 50 μg mRNA-LNP encoding rituximab. A total of five injections were administered until day 18 with two injections per week. In a follow-up study, mice received a total of six injections until day 21 either with 50 μg mRNA-LNP encoding rituximab or anti-influenza mAb or 200 μg recombinant rituximab with two injections per week. In both studies, animals were randomized using matched pair distribution method, based on body weight. Animals were checked daily for

**The paper explained**

**Problem**

While antibodies offer a potent and widely used means in cancer treatment, there are only three approved monoclonal antibodies against infectious diseases today, although protective antibodies have been described for a variety of pathogens. Thus far, antibodies are at a disadvantage compared to antibiotics and vaccines, predominantly for economic reasons. Hence, there is a strong demand for cost-effective alternatives to the administration of recombinant proteins. DNA-based antibody expression proved to reach protective serum titers in various preclinical models. However, permanent or at least long-lasting expression is often undesired for passive immunization, thus requiring sophisticated approaches for the clinic. By contrast, transience is inherent to another nucleic acid, mRNA. Moreover, a number of recent intriguing studies principally suggested mRNA as a means for molecular therapies requiring the delivery of functional proteins.

**Results**

*In vitro* and *in vivo* experiments revealed efficient expression of functional antibodies from exogenous mRNA. Flexibility of the technology was demonstrated by the expression of different antibodies and antibody formats such as classical IgG-based monoclonal antibodies (mAbs) or heavy-chain-only neutralizing agents (VNAs). *In vivo*, LNP–mRNA-mediated antibody expression protected mice against diverse biological threats such as virus, toxin, and neoplastic cell growth. Protection was obtained in pre- and post-exposure treatment scenarios. mRNA revealed a favorable expression kinetics giving rise to immediately elevated antibody serum titers. As a consequence, the efficacy of mRNA was comparable to recombinant antibody therapy in a particularly challenging post-exposure scenario for intoxication.

**Impact**

Today, passive immunization only fills a niche in preventing or fighting infectious diseases, mainly due to non-competitive costs compared to antibiotics and vaccines. However, there is renewed interest in passive immunization, for instance since the emergence of microbial resistance to antibiotics increased the demand for alternative therapies. The present study reveals that mRNA-mediated antibody expression can confer protection against diverse biological threats. We suggest formulated mRNA as a novel armamentarium for the development of competitive passive immunization therapies.

clinical signs and whole-body luminescence imaging was carried out at indicated times using a PerkinElmer IVIS® XR imaging system. Animals were euthanized upon excessive (> 15%) body weight loss and/or hindlimb paralysis (humane endpoint). In all challenge studies, a standard commercial rodent diet and tap water were provided to the animals *ad libitum*.

## Rabies VNT analyses

Mouse sera were analyzed for VNTs by a fluorescence antibody neutralization assay (FAVN assay) by EuroVir GmbH in Luckenwalde, Germany.

## HIV neutralization assays

BHK cells were transfected with VRC01 mRNA, and supernatants were subjected to a Magi R5-Tropic Antiviral Assay performed at Southern Research (Maryland, USA). The procedure of the assay has been described previously (Lackman-Smith *et al*, 2008).

Recombinant anti-HIV-1 gp120 monoclonal antibody (VRC01) and the CCR5 antagonist TAK779 were obtained from NIH AIDS Reagent Program (Cat. 12033 and 4983). Concentrations of VRC01 in supernatants were determined by an IgG-specific ELISA.

### Computational analysis

Data were analyzed using GraphPad Prism software version 6. All error bars depicted in this publication refer to standard deviations. ELISA data were analyzed using nonlinear regression. FACS plots were interpreted by BD FACSDiva™ software version 6.1.3 (Version), and data were further processed by *FlowJo*™ software version 10.0.7. Cells were gated on all living cells, and median PE fluorescence was calculated. Whole image processing was carried out using LI-COR's Image Studio version 5.2.5 according to manufacturer's recommendations. For correlation analysis of IgG titers and mouse anti-SO57 antibodies, a two-tailed nonparametric spearman correlation with a confidence interval of 95% was used.

**Expanded View** for this article is available online.

### Acknowledgements

We are very grateful to A. Urbschat, I. Biermann, T. Dam, A. Teegler, R. Betten, A. Kulessa, M. Debatis, J. Tremblay, A. Foss, and S. Chapman-Bonofiglio for excellent technical assistance throughout the project. We further thank C. Lackman-Smith for fruitful discussion on HIV assays and A. Gast for excellent technical assistance with the MAGI R5 antiviral assays. We are grateful to L. de Waal, K. Stittelaar, G. van Amerongen, and S. Berndsen from Viroclinics for their great support in designing and for conducting *in vivo* rabies studies. We are grateful to H. Pflicke for the conduction of tumor studies. We thank U. Kruse for careful reading of the manuscript.

### Author contributions

MT, JM, KF, AT, NH, RH, MF-M, CBS, and TS designed research. MT, JM, MP, KF, AT, BLM, MJH, YKT, and CBS performed research. MT, JM, MP, KF, AT, RH, CBS, and TS analyzed data. MT prepared the figures. MT and TS wrote the manuscript.

### Conflict of interest

MT, MP, AT, KF, NH, RH, MF-M, and TS are employees of CureVac AG developing therapeutics based on sequence-engineered mRNA. BLM, MJH, and YKT are employees of Acuitas Therapeutics, a company providing delivery solutions for molecular therapeutics using lipid nanoparticles.

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
