## [Review Process File · EMBO Molecular Medicine]

mRNA mediates passive vaccination against infectious agents, toxins and tumors

Moritz Thran, Jean Mukherjee, Marion Pönisch, Katja Fiedler, Andreas Thess, Barbara L. Mui, Michael J. Hope, Ying K. Tam, Nigel Horscroft, Regina Heidenreich, Mariola Fotin-Mleczek, Charles B. Shoemaker, Thomas Schlake

Corresponding author: Thomas Schlake, CureVac AG

Review timeline:

Submission date:	08 February 2017
Editorial Decision:	14 March 2017
Revision received:	14 June 2017
Editorial Decision:	04 July 2017
Revision received:	12 July 2017
Accepted:	13 July 2017

Transaction Report:

Editor: Céline Carret

1st Editorial Decision

14 March 2017

Thank you for the submission of your manuscript to EMBO Molecular Medicine. We have now heard back from the three referees whom we asked to evaluate your manuscript. Although the referees find the study to be of potential interest, they also raise a number of concerns that need to be addressed in the next final version of your article.

You will see from the comments below that the study is found of interest and should be considered for publication in EMBO Molecular Medicine. However, all reviewers request additional information and details are missing. Particularly, referee 1 highlights inadequate controls in the lymphoma model, and referee 3 does not find the rabies challenges rigorous enough, and suggests repeating them. We believe that the study would be much stronger if you could address all concerns. As such, we will consider a revision of your manuscript if you can address the issues that have been raised within the time constraints outlined below. Please note that it is EMBO Molecular Medicine policy to allow only a single round of revision and that, as acceptance or rejection of the manuscript will depend on another round of review, your responses should be as complete as possible.

Please also contact us as soon as possible if similar work is published elsewhere. If other work is

published we may not be able to extend the revision period beyond three months.

I look forward to receiving your revised manuscript.

***** Reviewer's comments *****

Referee #1 (Remarks):

Thran et al. present a passive immunization approach based on antibody-encoding mRNA. The mRNA was enclosed in lipid nanoparticles for targeted delivery to the liver of mice, where antibody was expressed. The paper discusses antibody kinetics and shows that mRNA based antibody expression can confer protection in vivo against various challenges including rabies virus, botulinum toxin and a mouse lymphoma model. The study is an interesting pre-clinical proof of principle.

Some points that should be addressed/require clarification in the manuscript:

-) How long do hepatocytes produce antibody? How do the kinetics compare to mAb transfer?
-) Starting from day 14 post anti-rabies mRNA transfer, becoming more pronounced around day 18-21, a subset of mice in seemingly all cohorts shows a pronounced decrease in antibody titers. The authors state in the discussion that they could not detect anti-drug antibodies, but do not state by what method this was tested. Did the authors attempt a serum transfer from mice with decreased antibody titers and attempted the vaccination in the recipient mice? Was a capture ELISA with the anti-rabies antibody coated on a plate attempted?
-) How did the treatment regimen for the lymphoma model compare to regular application of mAb? A better control to untreated mice would have been the use of e.g. mRNA encoding an anti-influenza antibody.
-) The authors state that therapeutic treatment against botulinum toxin was done 2, 4 or 6 hours post intoxication and refer to figure 4F. However, the figure only mentions 6 hours. There are no additional curves for 2 and 4 hours in figure 4G either.
-) In the discussion the authors state that persistent antibody expression is often neither required nor desired. This statement should be changed. Especially in pandemic scenarios where antibody treatment could be used in a preventative setting, longer bioavailability is important.
-) How long were the antibody ELISA plates incubated with coating solution? At what temperature were the plates incubated? What was the coating volume?
-) Was the disease scoring performed in a blinded fashion? The scoring can be somewhat subjective. Was it performed by a single operator or confirmed independently by a second operator? If done in an unblinded fashion by a single operator it should be clearly stated in the methods.
-) The authors state that the antibody sequences were codon optimized, but not for which organism.

Referee #2 (Comments on Novelty/Model System):

The rabies mouse model used does not mimic the classical setting for PEP

Referee #2 (Remarks):

In this study, Thran and colleagues explored the possibility to deliver antibodies using chemically unmodified mRNA formulated in lipid nanoparticles. Authors produced anti-viral, anti-toxin and anti-tumor antibodies in vitro and in vivo showing that their delivery conferred prophylactic and

therapeutic efficacy in mice challenged with rabies virus or with botulinum toxin. In addition, as a further example of the potential of this technology Authors provided evidence of efficacy in a mouse model of lymphoma.

In general, the manuscript is well written and a considerable amount of experimental data is provided.

Major points:

- Page 4: mention that a mAb is now approved in India to replace RIGs (RAB1 mAb, granted for market authorization as Rabishield in India in 2016). Rabishield is produced by the Serum Institute of India in partnership with MassBiologics USA. The product is being launched on the Indian market this year (Sloan et al. 2007 and Nagarajan et al, 2014)
- S057 in the cocktail named CL184 is well known in the field as CR57, it should be useful referring to it also as CR57.
- While "solitary" heavy chains are not secreted because retained in the ER, "solitary" free light chain could be secreted. Did the Authors check the release in vitro and in vivo of free light chains?
- Did the Authors assess the pro-inflammatory potential in vivo of the delivered LNP-mRNAs?
- The high variability in the survival rate of rabies virus challenged animals is rather unusual for such a highly lethal virus and might be related to the use of a low inoculum, can the Authors comment on the set up of the inoculum dose?
- The efficacy dose of 40 µg/mouse would translate into 2 mg/kg dose, is this dose feasibly deliverable in humans? Is this going to be cost-effective?
- While the rabies mouse model used in this study represents a valuable model to assess the antiviral efficacy of the LNP-mRNA, it is important to highlight that this is not a relevant model for rabies PEP. There are two important differences in the PEP model to mimic the treatment in humans of category 3 bites: animals are vaccinated and the RIGs are administered topically (at 20 IU/kg) to treat the wound site and to neutralize rabies virus while the active immune response to the vaccine is mounting.
- When the antibody is given topically, the local concentration of the antibody is considerably higher as compared to that measured in blood. Consequently, it appears that the i.v. administration of the antibody is suboptimal and that VNT blood levels of 10 IU/ml are not sufficient to protect from i.m. rabies virus challenge. Have the Authors considered the i.m. administration of the mRNA for this specific model?
- As in Figure 4G, it would be relevant comparing the efficacy of the LNP-mRNA delivered mAb with that of passively administered protein S057 mAb (given i.v. or topical).
- To consider as viable the proposed approach for mRNA-based rabies PEP, Authors should also consider that the efficacy of the rabies vaccine could potentially be reduced by high levels of vaccine-reacting pre-existing antibodies. Authors should demonstrate that the effective dose of 40 µg LNP-mRNA is not inhibiting the endogenous response to the vaccine. Interestingly, Authors have also developed up to Phase I a rabies mRNA vaccine and mention in the discussion the possibility of a co-administration of the vaccine and the Ab mRNAs. Have the Authors planned to prove this concept experimentally?

Minor points:

- Figure 1D: labeling of 3'UTR in the light chain is missing
- Figure 2B: indicate the concentration on the x-axis.
- Figure 2D: the staining of HA expressing cells appears modest if considering the binding to the mock transfectants, is the antibody reacting non-specifically to cell membranes?
- Figure 2E: Y-axis is likely mislabeled. The legends indicate that the plot shows the binding of mRNA produced Rituximab to Raji cells, is the y-axis showing fluorescence intensity as shown in the other panels of the same figure?

- Figure 3E: for consistency with the other panels the x-axis should be in days instead of hours.

Referee #3 (Remarks):

Thran et al report the use of unmodified mRNA to drive in vitro expression of well-known antibodies against rabies, influenza, HIV, and CD20, botulinum and Shiga toxin. Following successful expression in vitro, mRNAs encoding the antibodies against rabies, botulinum and CD20 were shown to be expressed in vivo when mRNA was delivered intravenously to mice in lipid nanoparticles. The expression levels were sufficient for passive immunization and showed full protection from challenge with botulinum toxin, partial protection from challenge with rabies virus and decelerated tumor growth caused by Raji lymphoma cells. Overall the paper is somehow novel and I believe it is suitable for publication in EMBO Molecular Medicine, however, there are several weak points that should be addressed.

1) While the use of unmodified mRNA for in vivo expression of antibodies is novel, there is plenty of literature showing similar experiments using DNA or viral vectors. The advantages of using mRNA described are not compelling. I believe this point should be discussed.

2) The levels of antibodies achieved in mice are sufficient to confer partial, or total, protection; however, similar levels of expression in humans would probably deliver extremely low concentration of antibodies, far from the protective threshold. This is a fundamental problem of this technology and is not addressed at all in the manuscript. The authors should discuss the limitations of this work and how this problem can be addressed.

3) Reference by Thess et al reported to support the notion that mRNA works in large animals is not appropriate. The reference reports mRNA driven expression of an hormone that is active in extremely low concentrations, while in the case of the antibodies reported in the manuscript require much higher concentrations.

4) The rabies challenges are confusing. It would be nice to repeat them and report a clean experiment.

1st Revision - authors' response

14 June 2017

Reviewer 1

-) How long do hepatocytes produce antibody? How do the kinetics compare to mAb transfer?

With relatively short-lived VNAs, in vivo data suggest that mRNA half-life contributes to the kinetics as discussed in the manuscript. However, the available data do not allow for any exact calculation. A very rough estimate with high uncertainties may indicate a half-life in the range of 12 to 18 hours. For conventional antibodies, it is impossible to deduce in vivo mRNA half-lives due to pronounced protein stability. Hence, we now addressed mRNA half-life in vitro, using primary mouse hepatocytes which may reflect the in vivo situation with liver as target organ most closely. We observed an mRNA half-life of about 15 h. In addition, we estimated the in vivo mRNA half-life of mRNA delivered to the liver via LNPs using short-lived erythropoietin. This analysis also suggests a half-life of roughly 15 h. This additional information was included in the manuscript.

Data from BALB/c mice suggest a serum half-life of recombinant Rituximab of about 8-10 days (about 10 days in human serum according to Suzuki et al., 2010) which is close to the estimated half-lives of mRNA-encoded anti-HA and anti-RAV-G antibodies measured in the present manuscript. Although these antibodies differ in detail, they all are humanized with (largely) identical Fc portions. Hence, it is reasonable (but not proven) to assume similar in vivo properties regarding their kinetics.

-) Starting from day 14 post anti-rabies mRNA transfer, becoming more pronounced around day 18-21, a subset of mice in seemingly all cohorts shows a pronounced decrease in antibody titers. The authors state in the discussion that they could not detect anti-drug antibodies, but do not state by what method this was tested. Did the authors attempt a serum transfer from mice with decreased

antibody titers and attempted the vaccination in the recipient mice? Was a capture ELISA with the anti-rabies antibody coated on a plate attempted?

Since SO57 was not available as recombinant protein, we initially used a different humanized antibody with the same Fc portion for coating of ELISA plates. With that approach we could not detect an ADA response correlating with in vivo antibody kinetics. Meanwhile, we produced recombinant SO57 antibody for ELISA plate coating. With that we now were able to detect a strong correlation between humoral anti-SO57 responses at the end of the study and the kinetics of antibody serum titers. We corrected the manuscript accordingly.

-) How did the treatment regimen for the lymphoma model compare to regular application of mAb? A better control to untreated mice would have been the use of e.g. mRNA encoding an anti-influenza antibody.

The treatment regimen was chosen based on a plenty of published mouse studies on the recombinant antibody. However, for better comparison we now conducted a further study using recombinant mAb as control. Moreover, we included the requested control with mRNA-encoded irrelevant antibody. Results of the new study were added to the manuscript.

-) The authors state that therapeutic treatment against botulinum toxin was done 2, 4 or 6 hours post intoxication and refer to figure 4F. However, the figure only mentions 6 hours. There are no additional curves for 2 and 4 hours in figure 4G either.

We agree that our wording was confusing and incorrect. While we did therapeutic treatment 2, 4 and 6 hours after intoxication, figure 4 only reveals the 6 h results. Data for 2 and 4 h are given in the supplementary material. We adapted the text accordingly.

-) In the discussion the authors state that persistent antibody expression is often neither required nor desired. This statement should be changed. Especially in pandemic scenarios where antibody treatment could be used in a preventative setting, longer bioavailability is important.

We agree that the wording was inappropriate and misleading. We adapted the text to be more specific. We now clearly talk about controllability as an advantage of using mRNA compared to DNA. Moreover, we now point out that there are also scenarios in which longer bioavailability is desired or even required. In those instances the use of DNA may be advantageous.

-) How long were the antibody ELISA plates incubated with coating solution? At what temperature were the plates incubated? What was the coating volume?

The requested information is now given in Material and Methods.

-) Was the disease scoring performed in a blinded fashion? The scoring can be somewhat subjective. Was it performed by a single operator or confirmed independently by a second operator? If done in an unblinded fashion by a single operator it should be clearly stated in the methods.

Treatment groups/cages were not blinded, but scoring was performed without knowledge of the treatment of the respective groups or symptoms on the previous day. Each scoring was performed by two operators. We included this information in Material and Methods.

-) The authors state that the antibody sequences were codon optimized, but not for which organism. Coding sequences were optimized for human use. It's now indicated in Materials and Methods.

Reviewer 2

Major points:

- Page 4: mention that a mAb is now approved in India to replace RIGs (RAB1 mAb, granted for market authorization as Rabishield in India in 2016). Rabishield is produced by the Serum Institute of India in partnership with MassBiologics USA. The product is being launched on the Indian market this year (Sloan et al. 2007 and Nagarajan et al, 2014)

The information was included in the manuscript as part of the introduction.

- SO57 in the cocktail named CLI84 is well known in the field as CR57, it should be useful referring to it also as CR57.

The information was included in the manuscript as part of the introduction.

- While "solitary" heavy chains are not secreted because retained in the ER, "solitary" free light chain could be secreted. Did the Authors check the release in vitro and in vivo of free light chains?

Figure S1 addressed the in vitro release of “solitary” heavy as well as “solitary” light chains. We do not have sufficient material and tools to investigate the same questions in vivo using the existing samples. However, conducting a new study applying mRNA-LNP encoding heavy or light chain individually would not be justified from an ethical perspective (we would not get it approved from the authorities). To reflect the in vivo situation more closely than before, we conducted an additional in vitro experiment using HepG2 cells. The results can be found as part of figure S1.

Just for explanation: expression levels are much lower in hepatocytes compared to BHK cells, since in vitro transfection efficacy of the former is rather poor.

- Did the Authors assess the pro-inflammatory potential in vivo of the delivered LNP-mRNAs?

Yes, tolerability of IV administration of mRNA-LNP was analyzed. All treatment regimens were well tolerated. Analysis of systemic cytokine levels revealed a transient weak increase for some proteins, which apparently did not hamper high protein expression. As liver is the target organ of mRNA-LNPs, histopathology was conducted on liver sections. The tissue revealed no signs of abnormality, including inflammation. To address the raised point, we included the respective data in the manuscript.

- The high variability in the survival rate of rabies virus challenged animals is rather unusual for such a highly lethal virus and might be related to the use of a low inoculum, can the Authors comment on the set up of the inoculum dose?

To conduct IM challenge experiments, an almost undiluted virus stock was used. Consequently, a substantial increase of the inoculum dose would have been impossible. It appears that IM infection is not very effective in mice. However, since the IM route reflects the natural situation most closely, it was chosen for the present study.

We agree that a “suboptimal” dose may have contributed to variability. However, there is no doubt that mRNA-encoded antibody can confer protection. Whether it is partial (since there may be some background survival) or complete is always also dependent on experimental parameters and from our perspective not crucial here. The manuscript’s intention is to demonstrate mRNA as a potential therapeutic option and not to provide a protocol ready for immediate transfer into clinical trials. In this respect please also note that most mRNA doses elicited VNTs well above the 0.5 IU threshold which is considered protective in humans. Whether this threshold also applies to mice is unknown. Our data suggest that it doesn’t, at least under the conditions used here.

Since variability does not correlate with the dose of control LNP, it is highly unlikely that unspecific effects of mRNA-LNP are responsible for partial survival.

- The efficacy dose of 40 µg/mouse would translate into 2 mg/kg dose, is this dose feasibly deliverable in humans? Is this going to be cost-effective?

Yes, in principle also the highest dose would be feasible in humans and would be cost-effective based on preliminary calculations. However, based on VNT data, 10 µg/mouse, i.e. 0.5 mg/kg, would be a more likely human dose for SO57, since it already elicited titers well above the accepted protective level.

- While the rabies mouse model used in this study represent a valuable model to assess the antiviral efficacy of the LNP-mRNA, it is important to highlight that this is not a relevant model for rabies PEP. There are two important differences in the PEP model to mimic the treatment in humans of

category 3 bites: animals are vaccinated and the RIGs are administered topically (at 20 IU/kg) to treat the wound site and to neutralize rabies virus while the active immune response to the vaccine is mounting.

We fully agree with the reviewer's comment. It was not the manuscript's goal to "perfectly" mimic PEP. Our work just shall suggest mRNA as a potential therapeutic option for passive immunization and not provide final solutions (neither in general nor specific to rabies). For instance, the treatment used here, IV administration, would not be competitive to existing rabies therapies. Therefore, using a "perfect" PEP model would not bring any benefit in the manuscript's context. To stress the fact that the manuscript shall provide PoC data instead of reflecting any real product development, we adapted the text.

- When the antibody is given topically, the local concentration of the antibody is considerably higher as compared to that measured in blood. Consequently, it appears that the i.v. administration of the antibody is suboptimal and that VNT blood levels of 10 IU/ml are not sufficient to protect from i.m rabies virus challenge. Have the Authors considered the i.m. administration of the mRNA for this specific model?

Clearly, to be competitive with existing rabies therapies, the mRNA approach would have to enable IM administration. This could lead to higher local concentrations, if good transfection efficiency can be achieved. However, we would not classify IV as suboptimal as such, since VNTs were protective according to human criteria with most doses.

In principle, LNPs can also be used for the IM route. However, they were developed for delivery to the liver. According to the respective MoA, transfection of muscle may be suboptimal. Hence, other formulations may have to be considered. In order to not further increase complexity, we did not address the IM route in the present manuscript and just included some theoretical thoughts in the discussion. We adapted the text to clearly indicate that our work does not provide a ready-to-use solution for IM.

- As in Figure 4G, it would be relevant comparing the efficacy of the LNP-mRNA delivered mAb with that of passively administered protein S057 mAb (given i.v. or topical).

We principally agree. However, since we had no access to pure mAb preparations enabling in vivo administration, such a control was not possible. This does not compromise the overall conclusion that mRNA gives rise to functional antibodies, in vitro as well as in vivo, which can confer protection. Analyses for VNAs and anti-HIV gp120 mAb (figure 2) demonstrate that recombinant protein and antibody expressed by mRNA do not differ in their functionality/efficacy. We don't see a rationale why this should be different for SO57.

- To consider as viable the proposed approach for mRNA-based rabies PEP, Authors should also consider that the efficacy of the rabies vaccine could potentially be reduced by high levels of vaccine-reacting pre-existing antibodies. Authors should demonstrate that the effective dose of 40 µg LNP-mRNA is not inhibiting the endogenous response to the vaccine. Interestingly, Authors have also developed up to Phase I a rabies mRNA vaccine ad mention in the discussion the possibility of a co-administration of the vaccine and the Ab mRNAs. Have the Authors planned to prove this concept experimentally?

As outlined before, IM administration would be necessary to be competitive for rabies. Hence, the analysis of potential interference would be primarily relevant for that route. However, IM administration was beyond the scope of the present manuscript and, thus, just mentioned as an outlook in the discussion. We now stated in the manuscript that the IM route may (or may not) require further substantial development. We plan to address this issue in the future. As an example, we referred to De Benedictis et al., 2016 to support the view that also using the right antibody can help to overcome potential limitations.

(Unpublished data for reviewers not included in the Peer Review Process File)

Minor points:

- *Figure 1D: labeling of 3'UTR in the light chain is missing*

It has been corrected.

- *Figure 2B: indicate the concentration on the x-axis.*

The experiment shown in Fig. 2B was conducted to demonstrate that recombinant and mRNA-derived antibodies are (largely) equivalent as to their quality/functionality. As it turned out later, starting concentrations slightly differed erroneously (10 vs. \approx 11 nM, as indicated in the figure legend). Hence, indicating concentrations in the graph is not possible (unless using two different scales, which from our perspective would be confusing). The slight difference between concentrations lies within the precision limits of the assay. Thus, the conclusion that both antibody preparations are largely equivalent is not affected.

- *Figure 2D: the staining of HA expressing cells appears modest if considering the binding to the mock transfectants, is the antibody reacting non-specifically to cell membranes?*

The experiment was used to demonstrate antibody functionality and thus set up rather as a qualitative than as a quantitative assay. Hence, it was not optimized for HA-expression, resulting in overall low fluorescence after staining. However, background staining appeared to be normal (please compare Mock intensities between C and D), i.e., there are no indications for unspecific binding of the secondary antibody. In conclusion, it is rather weak specific staining (due to low antigen expression) than non-specific antibody binding that gave rise to low differences in fluorescence intensities.

- *Figure 2E: Y-axis is likely mislabeled. The legends indicate that the plot shows the binding of mRNA produced Rituximab to Raji cells, is the y-axis showing fluorescence intensity as shown in the other panels of the same figure?*

The labeling was correct. Since Rituximab is available as recombinant antibody in contrast to all other candidates, we could use it as reference for quantification. However, we agree that also here fluorescence intensity should be used for the sake of consistency. Hence, we adapted the labeling as suggested.

- *Figure 3E: for consistency with the other panels the x-axis should be in days instead of hours.*

We adapted it as suggested.

Reviewer 3

1) While the use of unmodified mRNA for in vivo expression of antibodies is novel, there is plenty of literature showing similar experiments using DNA or viral vectors. The advantages of using mRNA described are not compelling. I believe this point should be discussed.

There are disadvantages (duration) and advantages (safety, controllability) of mRNA compared to DNA. Both nucleic acids are in principle qualified for therapies, and details of the indication determine, if both or either of them are suited. We adapted the text to better elaborate on the issue.

2) The levels of antibodies achieved in mice are sufficient to confer partial, or total, protection; however, similar levels of expression in humans would probably deliver extremely low concentration of antibodies, far from the protective threshold. This is a fundamental problem of this technology and is not addressed at all in the manuscript. The authors should discuss the limitations of this work and how this problem can be addressed.

Applying the same dose as in mice will definitely yield very low antibody titers in humans. However, there is a good rationale for the assumption that results can be transferred on a mg/kg base (e.g., Thess et al.). 10-40 μ g/mouse translates to 0.5-2 mg/kg which appears to be a feasible dose in humans. Nevertheless, we cannot exclude any unexpected differences between mice and humans. Clearly, this has to be addressed in the future, and the manuscript does not provide final proof for

the applicability of mRNA-mediated passive vaccination in humans. To reflect this better, we used more conservative statements in the text. We also highlighted that for instance treatment of rabies in humans would require IM administration for which the manuscript does not offer a ready-to-use solution. In the discussion it is pointed out that mRNA optimization, antibody identity etc. can help to overcome potential limitations. It is now also highlighted that the feasibility of passive mRNA vaccination has to be finally proven on a case-by-case basis.

3) Reference by Thess et al reported to support the notion that mRNA works in large animals is not appropriate. The reference reports mRNA driven expression of an hormone that is active in extremely low concentrations, while in the case of the antibodies reported in the manuscript require much higher concentrations.

We fully agree that erythropoietin requires much lower levels for biological activity compared to antibodies. Thus, Thess et al. does not prove the applicability of mRNA-mediated passive vaccination in large animals or even humans. However, this publication suggests that mRNA-mediated effects can be translated from small rodents to large animals and even primates. Clearly, this has still to be demonstrated for antibodies. To make this clearer, we adapted the wording where appropriate.

4) The rabies challenges are confusing. It would be nice to repeat them and report a clean experiment.

We agree that variability of controls complicates the understanding of challenge results. However, this does not compromise the basic message that mRNA represents a potential therapeutic option for passive vaccination. Despite the variability/"confusion", there is no doubt that mRNA-encoded antibody can confer (some) protection. Please also note that most mRNA doses elicited VNTs well above the 0.5 IU threshold which is considered protective in humans.

To conduct IM challenge experiments, an almost undiluted virus stock was used. Consequently, a substantial increase of the inoculum dose would have been impossible. It appears that IM infection is not very effective in mice. However, since the IM route reflects the natural situation most closely, it was chosen for the present work. In summary, repeating the experiment would not have eliminated "confusion".

To address the comment, we changed the wording to improve clarity. Where appropriate, we made statements more conservative to stress the fact that the manuscript presents results that may be compelling but do not yet provide a guarantee for applicability in humans.

2nd Editorial Decision

04 July 2017

Thank you for the submission of your revised manuscript to EMBO Molecular Medicine. We have now received the enclosed reports from the referees that were asked to re-assess it. As you will see the reviewers are now fully supportive and I am pleased to inform you that we will be able to accept your manuscript pending the following final editorial amendments:

- 1) Please address the minor text change commented by referee 1. Please provide a letter INCLUDING the reviewer's reports and your detailed responses to their comments (as Word file).
- 2) Ethics: please provide details for ALL mice experiments: age, gender, genetic background, housing conditions etc

Please submit your revised manuscript within two weeks. I look forward to seeing a revised form of your manuscript as soon as possible.

***** Reviewer's comments *****

Referee #1 (Remarks):

The authors addressed all the points brought up in the initial review. Additional experiments were performed where necessary and appropriately reported. All necessary clarifications were added to the text.

Minor comment:

-) In the newly added figure 5F, the legend should state if the lines indicate means or geometric means and what the error bars indicate.

Referee #2 (Remarks):

Authors have fulfilled and answered to most of the raised criticisms and requests.

Referee #3 (Remarks):

The revised version is suitable for publication.

2nd Revision - authors' response

12 July 2017

Referee #1 (Remarks):

The authors addressed all the points brought up in the initial review. Additional experiments were performed where necessary and appropriately reported. All necessary clarifications were added to the text.

Minor comment:

-) In the newly added figure 5F, the legend should state if the lines indicate means or geometric means and what the error bars indicate.

We added the requested information to the legend of Fig. 5F as well as to any other legend where the respective details were still lacking.

Corresponding Author Name: Thomas Schlake

Journal Submitted to: Embo molecular medicine

Manuscript Number: EMM-2017-07678